# Will the Inclusion of Generated Data Amplify Bias Across Generations in Future Image Classification Models?

## Abstract

As synthetic data becomes increasingly common in training pipelines, an open question is whether repeated use of generated data amplifies or mitigates bias in downstream models. Prior work has examined bias within generative models, but the downstream classifier's fairness across multiple generations remains less understood. We study this phenomenon through a multi-generation simulation in which a generative model and classifier are re-trained iteratively on mixtures of real and synthetic data.

Across three datasets and multiple architectures, we find that synthetic data does *not* produce a uniform trend: bias can increase, decrease, or stabilize depending on dataset structure, generator fidelity, and the synthetic-to-real mixing ratio. In particular, high-fidelity generators and balanced subgroup representation tend to preserve or reduce bias, while low-fidelity or imbalanced synthetic data can amplify bias over generations.

These results highlight that the impact of synthetic data on downstream fairness is highly context-dependent and cannot be characterized by a single monotonic effect. Our findings provide concrete conditions under which synthetic augmentation is likely to be safe and cases where it may gradually worsen model bias.

## 1 Introduction

As models continue to evolve and become more sophisticated, the demand for large amounts of high-quality training data has escalated (Alzubaidi et al., 2023). Traditionally, web data has been the primary resource for enhancing model performance (Deng et al., 2024). However, as this source becomes fully exploited, researchers have begun to explore alternative methods. One promising approach is to leverage generative models to create synthetic data (Fan et al., 2024; Meng et al., 2022; Zhou et al., 2023; Yang et al., 2023; Yeung et al., 2024; Alemohammad et al., 2023), thereby fueling continuous training cycles, as shown in fig. 1. This innovative self-sustaining pipeline effectively mitigates the issue of data scarcity, allowing models to improve iteratively with the help of their own generated outputs (Chen et al., 2024b; Lu et al., 2024). Despite the apparent advantages, this strategy introduces a crucial and complex debate: *Will the reliance on self-generated data eventually lead to model degradation*?

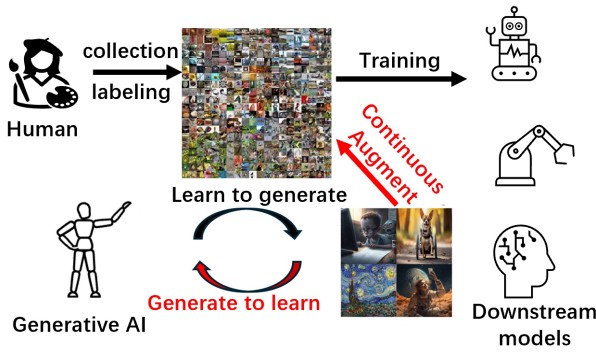

Figure 1: Generative models can be leveraged to generate more data to augment the training set, then help the downstream models training.

Some research has attempted to answer this question. On the one hand, Azizi et al. (2023) and Zhou et al. (2023) use the diffusion model to generate synthetic image data to augment the training set and observe the performance improvement in image classification tasks. Zheng et al. (2024) analyze the positive impact of generative data enhancement on small-scale datasets from a theoretical perspective. Hammoud et al.

states that a carefully designed generated data augmentation strategy could be helpful to alleviate the long tail problem. On the other hand, Alemohammad et al. (2024) show that purely adding the generated data to agent training could eventually cause model degradation, with their quality or diversity progressively decreasing. Singh et al. (2024) demonstrate that the use of synthetic data could cause a large performance drop in model robustness. The debate is still ongoing and remains unsettled.

It is important to note that although many research efforts are put into analyzing the influence of generative data on overall model performance, few of them explore the impact of generative data on the model bias, especially on the model's behavior in the worst-performing subgroups. Previous work (Zhang et al., 2024) has identified that models often behave significantly differently across various unknown subgroups, showing the critical role of model fairness in real-world applications. In the context of generative data, we raise a new question in this paper: will the inclusion of generated data help alleviate the model bias problem, or could it potentially make it worse? This question and its answer are significantly connected with other bias issues, *e.g.*, demographic parity (Loukas & Chung, 2023), equalized odds (Grant, 2023), maximum disparity (Roh et al., 2020), spurious correlation (Seo et al., 2022).

Intuitively, the bias issue is probably to be amplified because generated data is increasingly leveraged in training models across successive generations. Previous findings (Sehwag et al., 2022; He et al., 2024) reveal that generative models tend to sample data from high-density regions, leaving low-density data heavily under-explored. This imbalanced sampling introduces a natural skew in the dataset used to augment training, thereby exacerbating the bias present in the model. However, is this assumption accurate? To the best of our knowledge, no research has thoroughly explored this question. This lack of exploration leaves a critical issue unresolved, potentially creating an unknown risk in practical applications.

In this work, we study the impact of generated data on the model bias through the lens of the image classification problem, one of the most fundamental tasks of computer vision and deep learning. Our approach differs from previous studies in two key aspects: First, we focus on the impact of generated data on the model bias. Second, we create a more practical simulation environment by building a self-consuming loop that trains the generative model and the image classification model synergistically. We conduct experiments on three datasets, including colorized MNIST (Kim et al., 2019a), CIFAR-20/100 (Zhang et al., 2024), and Hard ImageNet (Moayeri et al., 2022b), to observe and analyze changes in various fairness metrics.

We summarize our contributions and key findings as follows:

1. We design and implement a scalable, self-consuming simulation environment. Our method interleaves dataset augmentation and model training across different generations.

2. We introduce data stacking and expert-guided filtering approaches to overcome data explosion and inconsistent data quality issues.

3. We conduct extensive experiments on three popular datasets to examine and reveal the impact of cross-generation generated data on model performance and bias.

4. We systematically analyze the factors causing diverse model bias behaviors.

## 2 Related work

### 2.1 Generative model and its application

Generative models have become a cornerstone of modern machine learning, particularly in the domain of data augmentation and synthetic data generation (Akkem et al., 2024). Early approaches, such as Generative Adversarial Networks (GANs) (Goodfellow et al., 2020), revolutionized the field by enabling the creation of highly realistic synthetic data through a process of adversarial training between a generator and a discriminator. More recently, diffusion models (Croitoru et al., 2023) have gained prominence due to their ability to generate high-quality data through a denoising process, offering an alternative to traditional GAN-based approaches. These generative models have been widely adopted in various tasks, including image synthesis (Liao et al., 2020), text generation (Li et al., 2018), and data augmentation (Antoniou et al., 2017),

proving their efficacy in improving model performance. In this work, we leverage two generative models, including the conditional GAN and text-to-image diffusion.

The advent of generative models has significantly expanded the possibilities for data augmentation by enabling the creation of entirely new data samples that mimic the distribution of the original dataset. For instance, Azizi et al. (2023) and Zhou et al. (2023) leverage diffusion models to generate synthetic images, successfully augmenting training sets and improving classification accuracy. Similarly, Zheng et al. (2024) explore the theoretical underpinnings of generative data augmentation, particularly in the context of small-scale datasets. However, the impact of using synthetic data is not without its challenges. Alemohammad et al. (2024) highlight that indiscriminate inclusion of generated data in training can lead to model degradation, where the model's performance deteriorates as the quality and diversity of the generated data decrease over time. Hammoud et al. observe a related phenomenon, noting that a carefully designed strategy for data augmentation could mitigate issues such as the long-tail problem. Further, Singh et al. (2024) demonstrate that the use of synthetic data can significantly undermine model robustness, leading to performance drops.

## 2.2 Bias in deep learning models

Many efforts have been made on the model bias. Kotek et al. (2023) investigate the behavior of large language models on gender bias. Liu et al. (2022) measure the political bias in language models. Zhang et al. (2024) identify the existence of subgroup bias in image classifiers. Hosseini et al. (2018) find the shape bias learning by convolutional neural networks. Khayatkhoei & Elgammal (2022) discover generative models can easily learn the spatial bias from the data. Heinert et al. (2024) and Hönig et al. (2024) research on texture bias in deep learning models.

There are also many fairness metrics to help evaluate bias (Kim et al., 2019b; Lin et al., 2022). Important fairness metrics include demographic parity (Jiang et al., 2022), which ensures that positive classification rates are equal across different demographic groups, and equalized odds (Romano et al., 2020), which requires that true positive and false positive rates are consistent across groups. Equal opportunity (Wang et al., 2023) further emphasizes equal true positive rates, ensuring that no group is disadvantaged in correct classifications.

## 2.3 Relation to Prior Work on Feedback Loops and Bias Propagation

Recent studies have examined how synthetic data or model predictions can contaminate future training distributions, producing feedback loops that amplify dataset biases. Taori et al. Taori & Hashimoto (2023) demonstrate that model-driven sampling can reinforce spurious correlations in the underlying dataset, resulting in a positive-feedback mechanism that progressively shifts the data distribution. Hataya et al. Hataya et al. (2023) analyze the long-term risks of large-scale generative models, showing that synthetic data can distort future datasets when used for uncontrolled large-scale scraping or augmentation. Chen et al. Chen et al. (2024a) further investigate whether deep generative models amplify bias in their own outputs, focusing primarily on bias internal to the generator.

Our work differs from these studies in several key ways. First, rather than analyzing bias amplification *within the generator*, we focus on how synthetic data affects the *downstream classifier*'s fairness across multiple generations of retraining. This distinction is important because downstream classifiers may respond to synthetic-data drift in ways that differ from the generative model itself. Second, we operationalize a multi-generation *interleaved loop* in which both the classifier and generator are retrained at each cycle, whereas prior work typically examines a single-component loop (either a generator feeding itself or a model influencing data collection). Third, we systematically measure multiple fairness metrics—including multi-class subgroup disparities and robustness against spurious correlations—allowing us to capture a broader spectrum of bias behaviors than binary-sensitive-attribute studies. Finally, our framework examines how generator fidelity, classifier capacity, and synthetic-to-real mixing ratios jointly shape the resulting bias dynamics, providing a unified view across datasets and architectures.

In summary, while prior work has shown that synthetic data can create harmful feedback loops, our contribution lies in characterizing how these loops affect *downstream fairness* in multi-generation training pipelines

and in identifying the conditions under which synthetic augmentation mitigates, preserves, or amplifies classifier bias.

# 3 Generate to Learn: Building a Scalable and Self-Sustaining Simulation Environment

## 3.1 A Simple yet Practical Framework for Simulation

To better understand the impact of the generated data on future model training, we design and implement a simulation environment grounded in real-world practices. The environment comprises four core components: subgroup construction, base model initialization, dataset augmentation, and future model development.

○ *Subgroup Construction.* Our environment is designed to study the effects of generated data on model bias, making it essential to establish clear and practical attributes for bias evaluation. Inspired by Zhang et al. (2024), we manually partition the original dataset into multiple subgroups, where subgroups within the same class share similar semantics. The introduction of bias is controlled by adjusting these subgroup partitions. During the training process, the models remain unaware of the subgroup partitions, which are only revealed during the evaluation stage to assess model bias.

○ *Base Model Initialization.* We construct and randomly initialize a base generative model $g(\cdot)$. This model is then trained from scratch on the dataset $\mathcal{D} = \{(x_i, y_i)\}_{i=1}^{N}$, where $x$ represents the sample to generate, $y$ is the corresponding label, and $N$ is the number of training samples in $\mathcal{D}$. The model is trained until it converges sufficiently on $\mathcal{D}$.

○ *Dataset Augmentation.* Once the base model is initialized, we use the generative model $g(\cdot)$ to generate data that approximates the distribution of the training dataset $\mathcal{D}$, thereby augmenting the original dataset. Because previous study (Zheng et al., 2024) has shown that training exclusively on generated data can eventually cause model failures, we adopt an alternative strategy (Azizi et al., 2023; Zhou et al., 2023), mixing the original data with generated data at a ratio of $p\%$.

○ *Future Model Development.* In addition to the base generative models, our task involves two types of models: downstream models and subsequent generative models. The downstream model corresponds to an image classification model optimized with cross-entropy loss. The generative model is a re-initialized version of $g(\cdot)$, trained on the augmented dataset from the previous generation. Unlike previous similar work that considers only a single generation, we incorporate generated data from multiple continuous generations, creating a more realistic and practical scenario.

We leverage the above core components to build our simulation environment. We begin with *subgroup construction* to study the model behaviors of interest. At each generation, we *(re)initialize the base model* using the current dataset, which may have been augmented. This model is then used to generate additional data for *dataset augmentation*. Finally, the *downstream models are developed* on the dataset augmented by the current-generation generative model.

## 3.2 Scaling the Simulation for Real-World Practice

Two significant challenges remain in our environment, limiting its scalability for simulating real-world practices: 1) *Data Explosion*: As the number of generations increases, the volume of generated data grows continuously, leading to a significantly larger training set and resulting in unbearable training time consumption. 2) *Inconsistent Data Quality*: Due to the inherent uncertainty in the generative process, the quality of data produced by the generative model across different generations may vary, potentially leading to degradation in the performance of future models.

We propose two strategies to incorporate into our simulation environment to address these challenges, including the *Data Stacking* and *Expert-guided Filtering*.

○ *Data Stacking.* We maintain a first-in-first-out queue to store the generated data. Specifically, we set the capacity of the queue to $D$. We continuously use the updated generative model to generate data with a

volume of $S$ and fill the queue until it reaches capacity, *i.e.*, the maximum number of generations that can be accommodated is $D/S$. Once the queue is full, the oldest data will be removed to make space for newly generated data.

○ *Expert-Guided Filtering.* We introduce two expert-guided strategies to filter low-quality samples and improve the quality of the training set. The first strategy involves conducting a human study to score the generated samples and removing those that are easily recognized as generated content. The second strategy leverages the CLIP and our trained classification model of the last generation to score the generated samples based on the prediction uncertainty (Gal & Ghahramani, 2016), filtering out the bottom $r\%$ based on their scores.

### 3.3 Evaluation Metrics for Assessing Model Bias

It is important to evaluate model performance, including bias, during the development process across generations. Consider an input $x \in \mathcal{X}$ from the initial meta training dataset in our simulation environment, associated with a ground-truth label $y \in \mathcal{Y}$. Assume the dataset comprises $L$ distinct classes, so $\mathcal{Y} = 1, 2, \ldots, L$. We hypothesize that each class is further divided into $G$ subgroups, assuming for simplicity that each class contains an equal number of subgroups, resulting in a total of $L \times G$ subgroups across the dataset. For each input $x$, its subgroup membership is denoted by $g \in 1, 2, \ldots, G$. The existence of such unknown

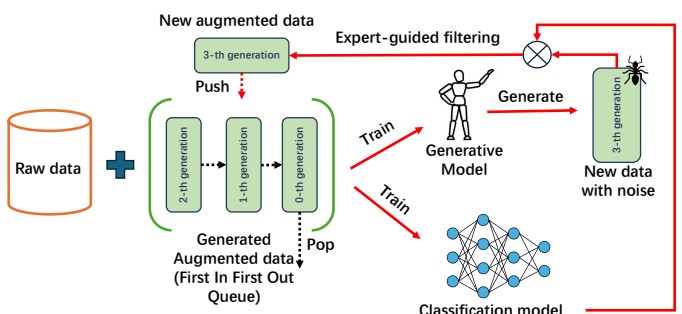

Figure 2: We continuously leverage new generators to produce additional images that enhance the training process, employing data stacking and expert-guided filtering to maintain high quality. We highlight the trajectory of the self-consuming loop in red.

subgroups and the varying model performance across these subgroups contribute to the presence of model bias. In our environment, we use several criteria to evaluate model performance, including overall performance, multi-group equality of opportunity, multi-group disparate impact, maximum disparity, and subpopulation performance. Among these metrics, we extend the conventional equality of opportunity and disparate impact to the content of image classification task with multiple tasks and multiple unknown attributes.

*Overall Performance.* We use the Fréchet inception distance (FID) (Heusel et al., 2017) and the classification accuracy (Acc) as metrics for the evaluation of the generative model and the downstream classification model.

$$\text{FID}_n = \|\mu_c - \mu_g\|_2^2 + \text{Tr}(\Sigma_c + \Sigma_g - 2(\Sigma_c\Sigma_g)^{\frac{1}{2}}), \quad \text{Acc}_n = P\left(f_n\left(x\right) = y_x\right), \tag{1}$$

where $\mu_c$ and $\Sigma_c$ are the mean and variance matrix of the feature vector extracted from Inception-V3 (Szegedy et al., 2015) on the original clean samples, $\mu_g$ and $\Sigma_g$ are those from the generated samples, $y_x$ is the ground-truth associated with the sample $x$, and $n$ indicates the number of generation.

*Equality of Opportunity.* The equality of opportunity (Ferreira & Peragine, 2013) measures whether every subgroup is treated equally by the model under study. In our simulation environment, we compute the equality of opportunity (EO) under the background of multiple groups as follows:

$$\text{EO}_n = 1 - \frac{1}{\binom{G}{2}} \sum_{i,j<G, i\neq j} \left\|\text{TPR}_n^i - \text{TPR}_n^j\right\|, \tag{2}$$

where we denote $\text{TPR}_n^i$ as the the true positive rate of $i$-th subgroup in the $n$-th generation. It can be computed as $\text{TPR}_n^i = P\left(f_n(x) = y \mid y = y_x, g = i\right)$, indicating the probability that the model $f_n$ correctly classifies an input $x$ from the $i$-th subgroup with the ground-truth label $y = y_x$.

*Disparate Impact.* Disparate impact (Feldman et al., 2015) measures whether different subgroups receive positive outcomes at similar rates. In our simulation environment, we extend this concept to multiple groups,

defining the multi-group disparate impact (DI) as follows:

$$\text{DI}_n = 1 - \frac{1}{\binom{G}{2}} \sum_{i,j<G, i\neq j} \left\| \frac{P(f_n(\boldsymbol{x}) = y_x \mid g = i)}{P(f_n(\boldsymbol{x}) = y_x \mid g = j)} - 1 \right\|, \tag{3}$$

where $P(f_n(\boldsymbol{x}) = y_x \mid g = i)$ denotes the probability that the model $f_n$ assigns a positive outcome (e.g., $y = y_x$) to an input $\boldsymbol{x}$ from the $i$-th subgroup.

*Maximum Disparity.* Maximum disparity measures the largest difference in model performance between any two subgroups. We compute the maximum disparity (MD) as follows:

$$\text{MD}_n = \max_{i,j<G, i\neq j} \left\| \text{TPR}_n^i - \text{TPR}_n^j \right\|. \tag{4}$$

*Subgroup Performance.* In addition to the aforementioned metrics for single-bias evaluation by pair-wise computation, we evaluate model performance by examining the accuracy of the multiple worst-performing subgroups. For each superclass $c$, we calculate the accuracy of its $G$ subgroups, denoted as $\text{Acc}_{c,g}$, and sort these accuracies in ascending order, $\text{Acc}_{c,(1)} \leq \text{Acc}_{c,(2)} \leq \cdots \leq \text{Acc}_{c,(G)}$. We then compute the average accuracy for each rank $k$ across all superclasses:

$$\overline{\text{Acc}}_{(k)} = \frac{1}{C} \sum_{c=1}^{C} \text{Acc}_{c,(k)}, \tag{5}$$

where $C$ is the total number of superclasses. This allows us to assess the model's performance across the most challenging subgroups.

Among these metrics, MEO (eq. (2)), DI (eq. (3)), and MD (eq. (4)) assess single-bias evaluation, and subgroup performance evaluates (eq. (5)) the impact of multiple biases.

**Why do we select these metrics?**  We do not choose to use the one-vs-rest (OvR) strategy (Jung et al., 2021) for evaluating fairness metrics in our multi-class classification tasks because OvR reduces multi-class problems to multiple binary subproblems, potentially missing the intricate biases and class interactions inherent in genuine multi-class contexts, thus overlooking unfairness arising from these interactions (Friedler et al., 2019). Additionally, OvR could introduce significant data imbalance in each binary subproblem, especially when class distributions vary greatly, which adversely affects classifier performance and distorts fairness metrics, leading to unreliable evaluations (Brzezinski et al., 2024). Instead, we employ fairness metrics specifically designed for multi-class classification — MEO, DI, and MD — to assess fairness across all classes simultaneously, preserving the integrity of the multi-class problem and providing a more accurate evaluation (Mazijn et al., 2021). This approach ensures that our fairness assessments reflect the complexities of multi-class classification, effectively manage potential data imbalances, and align with our objective to enhance fairness in a comprehensive and contextually appropriate manner.

## 4 Experiments

### 4.1 Evaluation setup

*Datasets.* We studied three datasets: Colorized MNIST, CIFAR-20/100, and Hard ImageNet. ① The Colorized MNIST dataset is a modified version of the original MNIST (LeCun, 1998), where three colors—red, blue, and green—are added to the images. We created two versions of this dataset. In the first, the three colors are uniformly applied across different classes. In the second, the colors are applied with uneven ratios, introducing a bias in the color distribution. ② The CIFAR-20/100 dataset is derived from CIFAR-100 (Alex, 2009) by grouping every five subclasses with similar semantic meaning into one single superclass, resulting in 20 classes. ③ Hard ImageNet (Moayeri et al., 2022b), a challenging subset of the ImageNet dataset(Deng et al., 2009), consists of 15 classes and contains various spurious correlations that can undermine the reliability of models trained on it.

*Models.* In the experiments with colorized MNIST and CIFAR-20/100, we consider five models: LeNet (Le-Cun et al., 1998), AlexNet (Krizhevsky et al., 2012), VGG-19 (Simonyan & Zisserman, 2014), ResNet-50 (He

Table 1: Evaluation of FID across generations for different generative models trained on various datasets, including colorized MNIST w/wo bias initialization, CIFAR-20/100, and Hard ImageNet. The 1st generative model is trained on the original dataset without the inclusion of generative data.

| Dataset | Initialization | Number of generations | | | | | | | | | |
|---|---|---|---|---|---|---|---|---|---|---|---|
| | | 1 | 2 | 3 | 4 | 5 | 6 | 7 | 8 | 9 | 10 |
| Colorized MNIST | Unbiased | 111.7 | 108.9 | 107.8 | 107.1 | 104.5 | 103.6 | 101.0 | 100.5 | 103.6 | 106.3 |
| | Biased | 109.4 | 106.0 | 107.03 | 106.4 | 105.3 | 114.2 | 109.0 | 108.6 | 109.1 | 116.5 |
| CIFAR-20/100 | N/A | 249.3 | 213.6 | 210.6 | 217.7 | 218.8 | 224.9 | 233.1 | 226.5 | 220.6 | 223.0 |
| Hard ImageNet | N/A | 56.6 | 49.9 | 55.4 | 60.2 | 70.6 | 65.8 | 153.2 | 256.1 | 353.1 | - |

et al., 2016), and MobileNet-V3 (Howard et al., 2019). For the Hard ImageNet experiment, we exclude the smallest model, LeNet, and additionally include a larger model, DeiT-S (Touvron et al., 2021). These models are sourced from the PyTorch library (Paszke et al., 2019), with the final layer modified to fit the specific classification tasks. We use GANs (Radford, 2015) to learn and generate the colorized MNIST and CIFAR-20/100 datasets, while stable-diffusion-1.5 (Rombach et al., 2022) is employed for generating the Hard ImageNet dataset.

*Metrics.* We evaluate model performance across all datasets based on classification accuracy. For the Colorized MNIST and CIFAR-20/100 datasets, which have explicit subgroups but are trained only at the superclass level, we also assess fairness metrics, including Multi-group Equality of Opportunity (MEO), Disparate Impact (DI), and Maximum Disparity (MD) (section 3.3). For Hard ImageNet, which contains spurious correlations without known subgroup partitions, we measure model accuracy on images with various ablation masks applied to the spurious objects.

*Implementations.* We set the number of generations to 10 or 4 in MNIST/CIFAR and Hard ImageNet, respectively. For training all models, we use the Adam optimizer, initializing the learning rate at $1 \times 10^{-1}$, with training capped at 50 epochs. Early stopping is employed to ensure full convergence and to avoid overfitting. We provide the evaluation of different generators across generations based on the FID score in table 1. The classification model at the 0-the generation is trained on the original dataset without any generated data. The queue has a maximum capacity of 3. For all results, We run 3 times to reduce the experimental randomness.

## 4.2 Evaluation on Colorized MNIST

We begin with the Colorized MNIST dataset, using both unbiased and biased initializations. The introduction of bias refers to the uneven painting strategy applied at the outset.

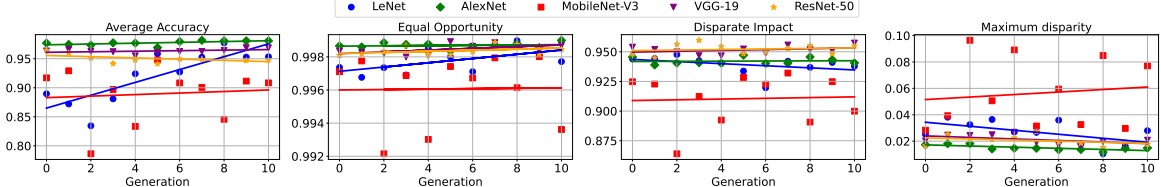

(a) Overall performance of the model trained on the Colorized MNIST dataset with unbiased initialization.

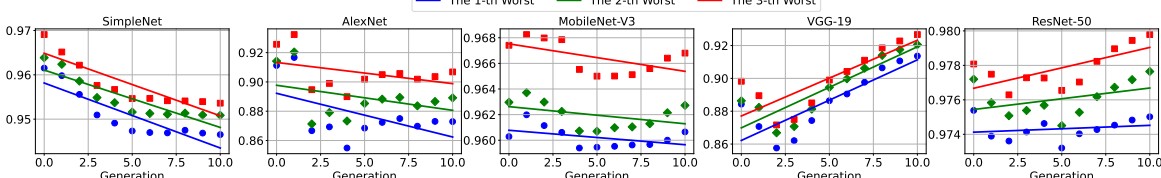

(b) Subgroup performance of the model trained on the Colorized MNIST dataset with unbiased initialization.

Figure 3: Results on the models trained on the MNIST dataset with unbiased initialization.

*Unbiased Initialization.* The results are shown in fig. 3 . Most models benefit from data augmentation using the updated generated data across generations, and all single-bias evaluations also show slight improvements. However, there are notable exceptions, particularly with MobileNet-V3, which experiences significant performance variations across generations. It's important to highlight that models differ considerably in multi-bias evaluations. While VGG-19 and ResNet-50 show significant improvements, smaller models, including SimpleNet, AlexNet, and MobileNet-V3, exhibit a noticeable decline in subgroup performance with continued large generations.

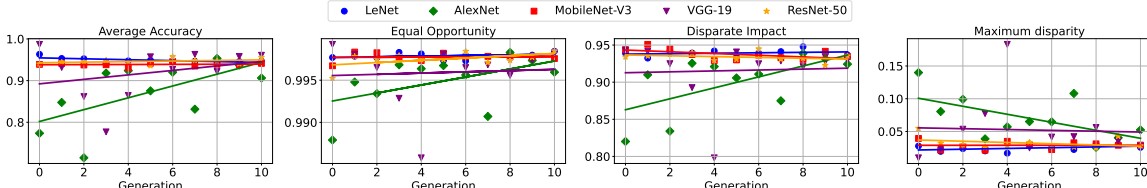

(a) Overall performance of the model trained on the Colorized MNIST dataset with biased initialization.

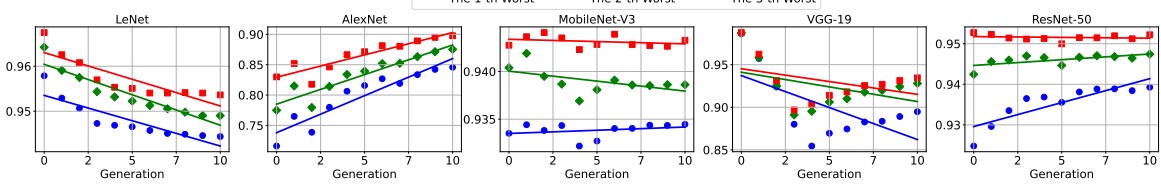

(b) Subgroup performance of the model trained on the Colorized MNIST dataset with biased initialization.

Figure 4: Results on the models trained on the MNIST dataset with biased initialization.

*Biased Initialization.* We present the results of models trained on the dataset with biased initialization in fig. 4. We observe consistent results in terms of classification accuracy and single-bias evaluation, which presents continuously imrprovement across generations; however, there are significant differences in the multi-bias evaluation. Specifically, VGG-19 experiences substantial performance degradation across subgroups, despite improvements on the dataset with unbiased initialization. In contrast, AlexNet performs better on this dataset as the number of generations increases. Compared with the results on the colorized MNIST with unbiased initialization, though MobileNet-V3 presents stable performance in this environment, both of the AlexNet and VGG-19 show large variation.

**Summarization & Takeaways**. As reported in table 1, the generative model can learn an approaching latent representation similar to that of the real samples on the colorized MNIST datasets, which is evident by the similar results on the FID evolution across generations. Thus, we can omit the impact of the quality of the generated data on the downstream models here. On the MNIST dataset, models can be consistently improved by augmenting the dataset with generated data across multiple generations. Notably, the inclusion of additional generated data does not significantly affect the models' single-bias performance, even with a large number of generations. However, it can lead to substantial variations in subgroup performance, revealing the presence of the multi-bias problem. The impact of generated data across generations varies between different models but remains consistent within the same architecture over multiple generations. Comparing results from unbiased and biased initializations, we observe that the presence of bias in the original dataset does not cause the model to degrade rapidly. Both initialization types exhibit similar trends in single- and multi-bias performance. In other words, the presence of dataset bias does not significantly amplify model bias when the dataset is augmented with generated data across generations.

## 4.3 Evaluation on CIFAR-20/100

Next, we proceed to a more challenging dataset, CIFAR-20/100. Different from MNIST, the original CIFAR dataset comprises more features and biases influencing the model training, which are not easily controllable. Thus, we investigate the impact of using pre-trained weights on the model bias during the self-consuming loop. In this experiment, we compare the performance of models initialized with pre-trained weights provided by the PyTorch library to those trained from scratch. This comparison will help assess the effectiveness

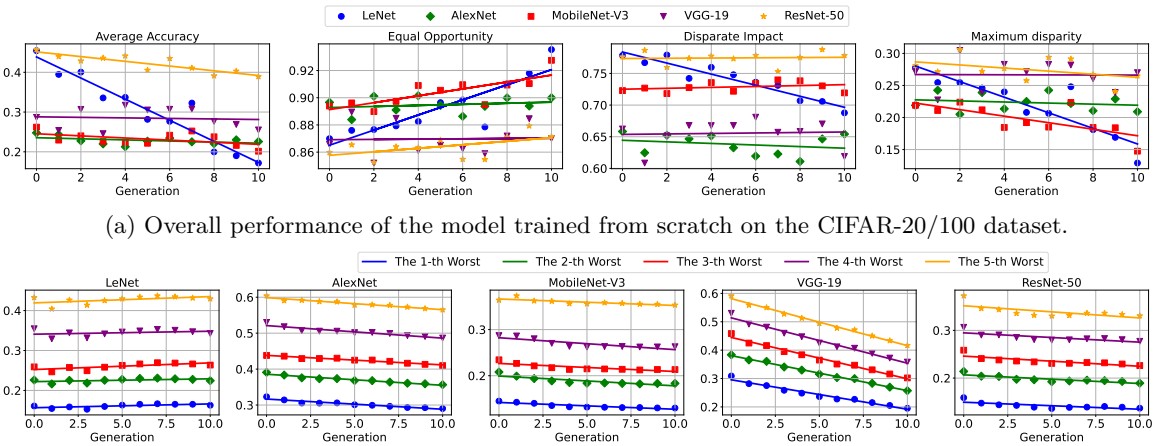

(a) Overall performance of the model trained from scratch on the CIFAR-20/100 dataset.

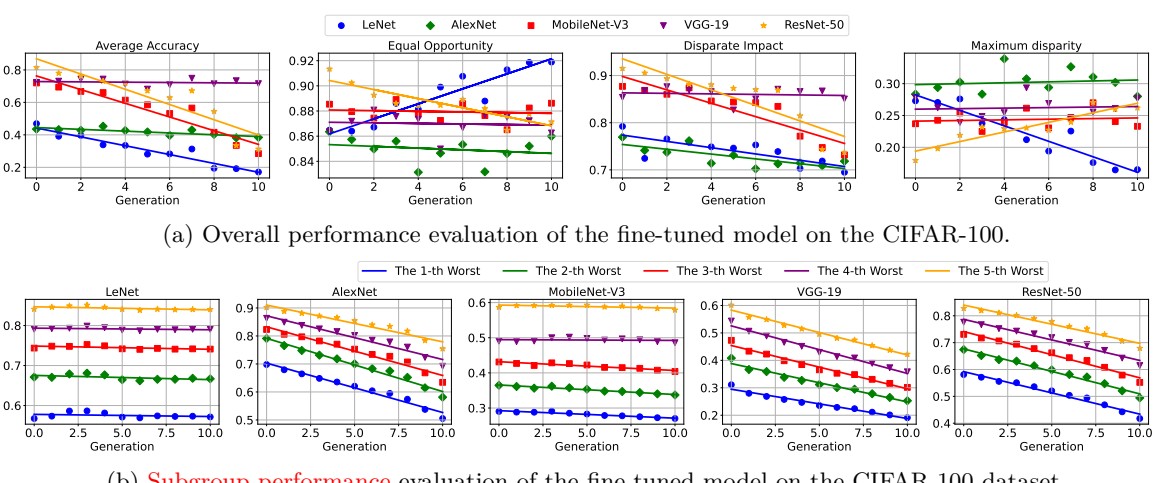

(b) Subgroup performance of the model trained from scratch on the CIFAR-20/100 dataset.

Figure 5: Results on the models trained from scratch on the CIFAR-20/100 dataset.

of pre-trained weights in improving model performance and stability when applied in this iterative data augmentation process.

*Without Pre-trained Weights.* Unlike the results on the MNIST dataset, augmenting CIFAR-20/100 with generated data can lead to degradation, with LeNet experiencing up to a 20% drop after 10 generations. The impact on bias metrics also varies. In the single-bias evaluation, both Equality of Opportunity and Maximum Disparity are significantly improved across all models, while most models show similar behavior regarding Disparate Impact. LeNet exhibits a larger bias in terms of Disparate Impact. For the multi-bias evaluation, models perform more consistently across different subgroups compared to their average performance across generations. Notably, although VGG-19 shows decreasing performance over generations, it performs better in bridging the performance gap between different subgroups.

(a) Overall performance evaluation of the fine-tuned model on the CIFAR-100.

(b) Subgroup performance evaluation of the fine-tuned model on the CIFAR-100 dataset.

Figure 6: Results on the models pre-trained on the ImageNet and fine-tuned on the CIFAR-20/100.

*With Pre-trained Weights.*Notably, models perform a faster performance degradation when using pre-trained weights as the number of generations for data augmentation increases. A greater number of models exhibit declines in classification accuracy and fairness metrics, such as Equality of Opportunity and Disparate Impact. Interestingly, while ResNet-50 without pretraining does not show a significant performance drop in the multi-bias evaluation, it experiences substantial degradation when pre-trained on the ImageNet dataset. This suggests that pre-trained weights, despite their initial advantage, may exacerbate model bias and performance issues in this iterative augmentation process.

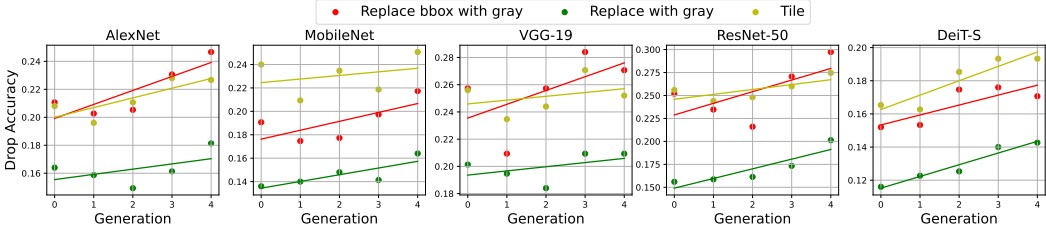

Figure 8: Evaluations of the impact of spurious correlation on the models pre-trained on the ImageNet and fine-tuned on the augmented Hard-ImageNet dataset across generations.

**Summarization & Takeaways**. As shown in table 1, continuously training on the dataset augmented by generated data across multiple generations leads to a slight improvement in generative performance, as evidenced by the decreasing FID scores on the CIFAR-20/100 dataset. However, despite the improved generative model, classification models trained with successively augmented datasets still experience a decline in performance in both the original classification task and bias evaluations. When using pre-trained weights from the ImageNet dataset, the classification models show significant improvement compared to training from scratch. Nevertheless, it is evident that models with pre-trained weights are more susceptible to integration bias introduced by the augmented datasets evolved over generations, further exacerbating performance deterioration in bias evaluations.

### 4.4 Evaluation on Hard ImageNet

We also conduct experiments on Hard Imagenet (Moayeri et al., 2022a), a dataset gathered from ImageNet with very strong spurious cues. The dataset contains 15 classes, and in each class, there is a strong correlation between the image background and the objects. This may lead the model to rely on background information rather than the actual objects for classification. Compared to the pre-defined color bias in Colorized MNIST and existing subgroup biases in the CIFAR-20/100 dataset, the unknown spurious correlation bias in this dataset is more challenging and difficult to fully identify, making it harder to mitigate during model development.

To study the impact of cross-generational data on this model bias, we made a modification to our proposed simulation framework. First, we fine-tuned the Stable Diffusion model using Low-Rank Adaptation rather than training from scratch to achieve a good balance between efficiency and generation quality on our task. Then we use 5 generations of mixed datasets to fine-tune our classifiers. Subsequently, while lacking explicit signal for single and multiple bias attributes, ablation studies are conducted on each classifier, following the approach described in Moayeri et al. (2022a). Specifically, we performed three types of ablation: (1) the object pixels were replaced with a uniform value of 0.5, neutralizing the object's appearance; (2) the entire bounding box surrounding the object was replaced with gray, removing shape-related information, and (3) the bounding box was replaced with a neighboring region of the image, substituting the object with local context. The performance drop caused by masking the image can indicate the model's reliance on spurious correlations. A significant performance drop suggests that the model's predictions rely more on the core object, indicating less influence from spurious correlations.

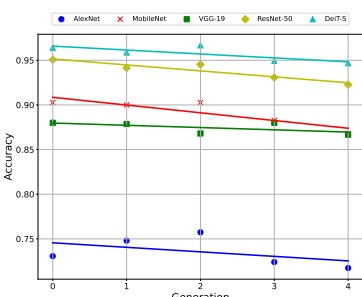

Figure 7: Subgroup accuracy of models fine-tuned on the augmented dataset across generations.

We report the changes in classification accuracy across generations in fig. 9 and the impact on learned spurious correlations in fig. 8. Over time, we observe that generated data can degrade model performance, as evidenced by the negative correlation between performance and the number of generations. However, for smaller models, performance sees a notable improvement during the first two generations, after which it stabilizes or becomes slightly better than the initial generation. Regarding bias evaluation, all models show a tendency to rely less on spurious correlations, indicating a shift toward focusing more on the core object for classification.

# 5 Why Models Exhibit Diverse Behaviors Across Generations

The varied behaviors observed across different datasets and models can be attributed to several factors, including the datasets, models, and data quality across generations. These factors interact with each other in complex ways, influencing the dynamics of bias across generations.

*Dataset Characteristics.* Different datasets exhibit unique features such as image complexity, class diversity, and inherent biases. Let $\beta_D$ represent the inherent bias in the dataset. For simpler datasets like Colorized MNIST, generative models can learn accurate representations more easily, resulting in generated data that closely matches the original data distribution. This closeness can be quantified by a high data quality factor $q_t \approx 1$ at generation $t$. The generated data with minimized bias helps the model continuously improve its classification performance and reduce bias.

*Model Architecture Sensitivity.* Different model architectures have varying capacities to learn from augmented data and mitigate bias. Let $\gamma_M$ represent the model's capacity to mitigate bias, which is a function of the model's architecture $M$. Larger models with higher capacity (e.g., VGG-19, ResNet-50) have higher $\gamma_M$, enabling them to handle biases in the data better. Conversely, smaller models (e.g., LeNet, AlexNet) have lower $\gamma_M$ and are more susceptible to biases in the training data, leading to greater performance variability across generations.

*Exposure of Bias.* Datasets contain various biases, both known and unknown, explicit or difficult to detect. The exposure of bias can be represented by a bias amplification factor $\delta$, which accounts for the complexity and ingrained biases within the dataset. As biases become more difficult to identify—progressing from color bias to subgroup bias and spurious correlations—we observe greater fluctuations in model performance. The bias in the model at generation $t + 1$, denoted $B_{\text{model}}^{(t+1)}$, can be influenced by the bias in the data and the model's capacity to mitigate it.

*Unbalanced Generation.* As identified in previous studies (Sehwag et al., 2022; Lee et al., 2021), generative models typically generate data from high-density regions of the data distribution, potentially over-representing certain classes or features. This tendency can be represented by an unbalanced generation factor $u_t$ at generation $t$, which contributes to the bias in the generated data. The quality of data generation is crucial; lower-quality data can degrade the overall representation quality, which may mitigate biased performance in downstream models by introducing noise.

Combining these factors, we have a conjecture about modeling the bias dynamics across generations using a recursive relationship. Let the bias in the model at generation $t + 1$ be expressed as:

$$B_{\text{model}}^{(t+1)} = (1 - \gamma_M)\left(1 + \delta_D + \delta_Q(1 - q_t) + \delta_U u_t\right) B_{\text{model}}^{(t)}. \tag{6}$$

Then, the overall bias amplification factor $A_t$ at the generation $t$ can be denoted as $A_t := (1 - \gamma_M)\left(1 + \delta_D + \delta_Q(1 - q_t) + \delta_U u_t\right)$. Depending on the values of $\gamma_M$, $q_t$, $u_t$, and the constants $\delta_D$, $\delta_Q$, and $\delta_U$, the bias amplification factor $A_t$ can be greater or less than 1. If $A_t > 1$, the bias increases across generations; if $A_t < 1$, the bias decreases.

Thus, the interplay between dataset characteristics, model architecture sensitivity, exposure of bias, and unbalanced generation may determine the bias dynamics across generations. To establish a self-sustaining model development loop with positive feedback, it is essential to have a clearer understanding of dataset bias ($\delta_D$), utilize larger models with higher capacity ($\gamma_M$), and employ high-quality generative models with improved sampling mechanisms to increase $q_t$ and reduce $u_t$.

# 6 Conclusion

Several models, like Stable Diffusion (Rombach et al., 2022), LLaMA (Touvron et al., 2023), LLaVA (Liu et al., 2024), and Nemotron (Adler et al., 2024), involve self-consumption loops. Notably, Nemotron is trained with over 98% synthetic data. While synthetic data can improve training, it may also introduce risks, particularly related to model biases. This has led us to investigate how generated data affects model

performance and bias, especially as self-consumption loops increase. Our experiments on Colorized MNIST, CIFAR-20/100, and Hard ImageNet datasets show that bias changes depend on factors like dataset type, model architecture, and generative model performance. Additionally, models are more sensitive to multiple biases than to a single one.

## 7 Limitations, Ethics, and Broader Impact

This work analyzes how synthetic data influences downstream fairness across multiple generations of training. The study relies on controlled datasets with simplified or constructed subgroup structures, which enable precise measurement of bias dynamics but do not fully reflect real-world demographic attributes. The generative and classifier architectures evaluated represent a subset of available model families, and different choices may exhibit distinct behaviors under self-consuming loops. Our simulation framework assumes synchronized retraining of generator and classifier, which captures one interpretable feedback regime but does not exhaust the variety of update patterns found in practical systems.

The ethical motivation of this work is to understand how synthetic data may amplify or mitigate bias as it propagates through iterative training pipelines. Although the experiments do not involve personal or sensitive attributes, the mechanisms uncovered here are relevant to fairness risks that arise when synthetic augmentation becomes part of large-scale data ecosystems. Misuse or unmonitored deployment of such pipelines could reinforce systematic biases if contamination effects are poorly understood.

By characterizing when generational contamination is likely to stabilize, worsen, or improve downstream fairness, this work provides insights that may support safer design of synthetic-data pipelines, more informed dataset curation practices, and improved governance tools for long-horizon training workflows. The analysis does not prescribe normative fairness criteria but aims to supply empirical principles that help develop more robust and accountable machine learning systems.

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

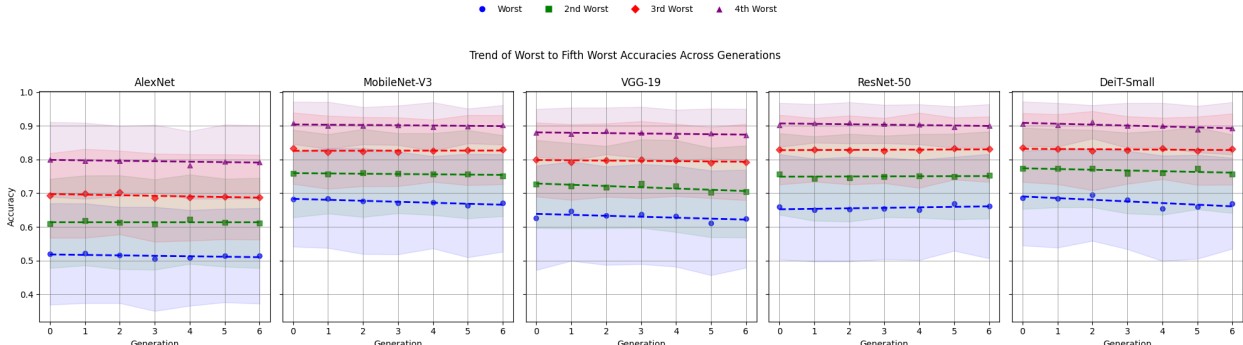

Figure 9: Classification accuracy of models fine-tuned on the augmented ImageNet across generations.

## A More results on the ImageNet

We conduct additional experiments on the ImageNet dataset Breeds, which is organized by subclasses as defined by WordNet. Each superclass consists of four subclasses. Following the same settings used for Hard-ImageNet, we utilize Stable Diffusion 1.5 to learn the dataset's distribution and augment it with generated data across multiple generations. In each generation, we train AlexNet, MobileNet-V3, VGG-19, ResNet-50, and DeiT-Small. The results are shown in fig. 7.

We observe a consistent phenomenon with the results on Hard-ImageNet. Compared to the best-performing subgroup, the generated data has a greater impact on the worst-performing subgroups, as indicated by a steeper slope across different generations.

## B Examples of generated images across generations

As shown in fig. 10, fig. 11, and fig. 12, each row represents a generation of images, with the generation number increasing sequentially from top to bottom. We can find that on MNIST and CIFAR-20/100 dataset, the quality of generated data doesn't change a lot, while it decreases significantly for the Hard ImageNet dataset.

## C Details on the expert-guided filtering

First, we manually review the generated samples and discard images with low quality.

Second, we calculate the CLIP score for each image, where the paired text is the class name. Images are then grouped into bins based on their CLIP scores, with each bin representing a $\pm 10\%$ range of CLIP scores. This results in 10 bins.

Then, we randomly sample 10 images from each bin and evaluate the quality of each bin. Based on this evaluation, we determine the maximum ratio of the CLIP score range (denoted as $r\%$) to retain for training.

- For MNIST, we find that retaining the top 90% of images ($r = 10\%$) is optimal.
- For CIFAR-20/100, retaining the top 70% of images ($r = 30\%$) works best.
- For the ImageNet dataset, retaining the top 40% of images ($r = 60\%$) yields the best results.

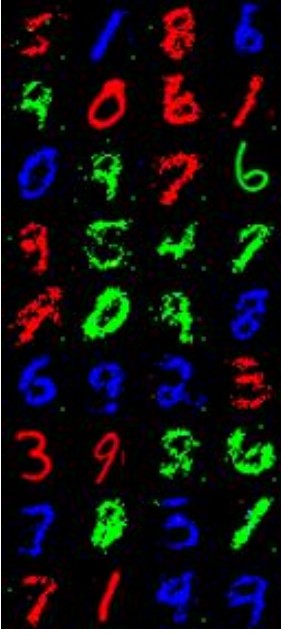

Figure 10: Color-MNIST

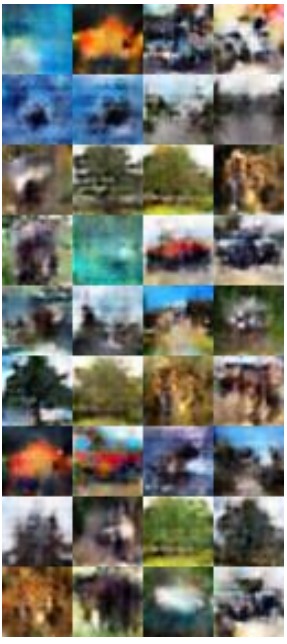

Figure 11: CIFAR-20/100

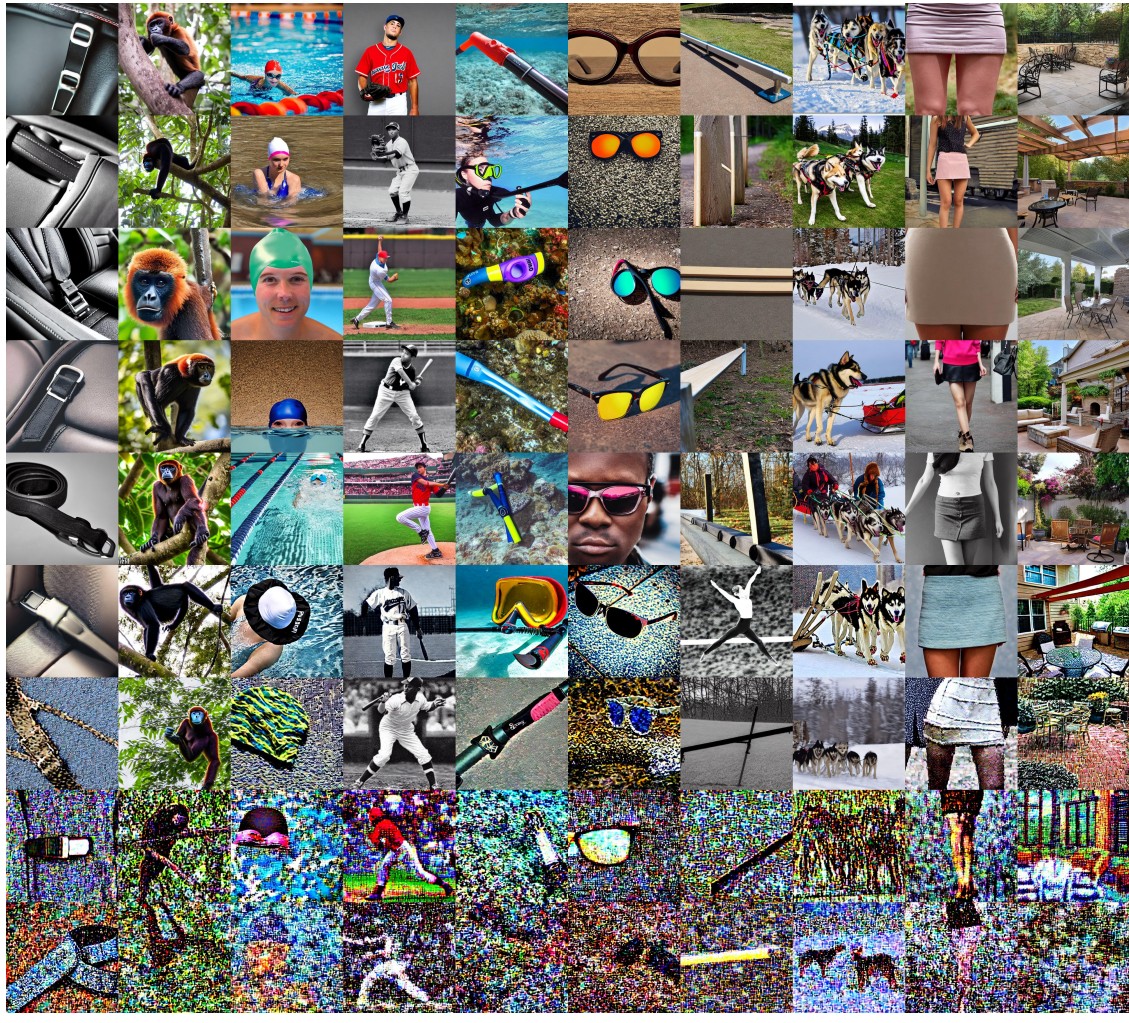

Figure 12: Hard ImageNet

