# OpenReview forum: "Will the Inclusion of Generated Data Amplify Bias Across Generations in Future Image Classification Models?"
_TMLR — Rejected by TMLR_

### Review · Reviewer_2ndw · 2025-09-29

**Summary Of Contributions:**

## Summary
This paper investigates the question of whether synthetic data will impact the (subgroup) fairness of models trained on this data, possibly combined with real data. For this, the paper proposes the following setup:
  1. Starting with a dataset of class-labeled natural images, train a generative model and a classifier on this data
  2. In multiple iterations, generate synthetic data using the generative model, and continue training the generative model on the real data + synthetic data of the 3 previous iterations.
  3. For each iteration, train a classification model on the real + synthetic data and observe its overall classification accuracy, in addition to fairness metrics such as Equality of Opportunity.

The paper trains multiple classification models of different architectures in this setup, using colored MNIST, CIFAR-20 and Hard-ImageNet as underlying real datasets. Findings are to some extent inconsistent: Quality of synthetic data (in terms of FID) stays almost constant over iterations for MNIST and CIFAR, but significantly deteriorates for Hard-ImageNet. Fairness metrics seem to be governed by model effects rather than data, as for some models, fairness increases over time, while it decreases or stays constant for other models.

## Strengths
**(S1)** The setup is very interesting and realistic. Evaluating the effects of data augmentation with synthetic data on accuracy, generation quality, and fairness over multiple iterations is very relevant and can yield interesting insights.

**(S2)** The experimental setup is comprehensive, involving multiple model architectures and datasets, including an ImageNet variant. It is not reasonable to expect more extensive experiments due to the computational cost associated with them.

**(S3)** The paper evaluates a number of fairness metrics, which are sufficient for a comprehensive assessment.

## Weaknesses
**(W1)** The main weakness of this paper is the missing essential information and the unclear exposition of the main research question.

Regarding the generative model: Is it conditional or unconditional, i.e. can we control the classes for which to generate images, or is this unconstrained? It would be important to know if there are any constraints on what the synthetic data contains, i.e. if it is enforced that synthetic data maintains the class (and subgroup) distribution of the real data, or not.

Regarding the synthetic data, how many images are generated in each iteration? Sec. 4.1 mentions "The queue has a maximum capacity of 3." I assume this means "3 iterations", but how many images is this?

Regarding the subgroups: What are the different subgroups in each dataset, and how many are there? Sec. 3.1 mentions "we manually partition the original dataset into multiple subgroups, where subgroups within the same class share similar semantics". This is not sufficient, given that subgroup bias is the main target of this paper.

Regarding model training, what is the fraction $p$ of real data? I didn't find this information in the paper. Additionally, it would be very interesting to see how different ratios of real vs. synthetic data influence fairness.

Regarding the different training iterations: How does the dataset composition change over time? Are the class and subgroup proportions constant over time, or can they change? This would be very important to know, and to assess if bias increase/decrease is moderated by over/underrepresentation in the synthetic data, or if only data quality/faithfulness is relevant.

Regarding the experimental setup, why is the experimental setup for Hard-ImageNet different from that of the other two datasets? Here, spurious correlation with background is evaluated, which is different from the other two datasets. I strongly recommend using a unified experimental setup and reporting additional, diverging experiments separately.

**(W2)** The role of Sec. 5 is unclear to me. How are the individual quantities measured exactly, and is it possible to connect the results from the experiments to Eq. 6 and derive concrete values for the variables?

**(W3)** In Sec. 3.3, the relation of justification of "Why do we select these metrics?" to the rest of the section is unclear to me. It mainly contains a rationale for why not to use a specific one-vs-rest metric, which seems less relevant for the paper, while arguments for multiclass fairness metrics are generic. Consider removing this part or moving it to the supplementary material.

**(W4)** Some references and citations need to be revised. For GANs, the 2024 NeurIPS paper by Goodfellow et al. should be cited [a]. Sec. 2.2 should cite [b] since it informs the main fairness metrics in this paper. Overall, the related work section could be expanded, as it is short and does not contain essential references such as [c] with already a similar title (and topic).

**(W5)** Regarding expert-guided filtering, it would be interesting to know the absolute number of filtered images, and a comparison to purely automatic filtering would be helpful. As this step implies significant human involvement and non-reproducibility, this significantly impacts the generalizability and scalability of insights.

**(W6)** Hyperparameters for training models are reported only partially. In addition, the paper mentions a learning rate of $1\times 10^{-1}$, which seems a very high learning rate value, possibly explaining the inferior quality of generated images in Fig. 10-12. Was any hyperparameter tuning performed to determine these?

**(W7)** As already mentioned, the quality of the synthetic images in Fig. 10-12 seems low, and I disagree with Appendix C that only synthetic images for Hard-ImageNet are of inferior quality. Images for CIFAR are very blurry, and many MNIST numbers are significantly corrupted as well. The main challenge this poses is that it becomes hard to disentangle effects on the classifiers stemming from more _biased_ synthetic data in comparison to more _noisy_ synthetic data.

**(W8)** There are formatting issues, such as typos and non-standard line breaks, for example, within the title of Sec. 5 and the top of p. 2. On page 7, consider floating all figures to the top, to avoid the 2 lines of main text being squeezed between 1 table and 2 figures, making the text hard to follow. Overall, please carefully revise the formatting and orthography of the entire paper.

## References
[a] Goodfellow et al.: Generative adversarial nets. In NeurIPS 2014.\
[b] Hardt et al.: Equality of opportunity in supervised learning. In NeurIPS 2016.\
[c] Chen et al.: Would Deep Generative Models Amplify Bias in Future Models? In CVPR 2024.

**Audience:**

Yes

**Audience Explanation:**

Findings how synthetic data impact classifiers (partially) trained on it has important implications for several areas of machine learning research.

**Broader Impact Concerns:**

Although this paper is on fairness, it does not contain a Broader Impact section, and it does not discuss any societal or ethical implications. However, this is required here due to the topic. This section should describe how the proposed method addresses and possibly mitigates societal harms caused by AI deployment, and additionally, it should critically discuss how the findings and methods proposed in this paper themselves could cause or contribute to such harms.

**Claims And Evidence:**

No

**Claims Explanation:**

While the overall setup and questions asked in this paper are interesting, as described in detail above, there are many details missing. Most importantly, the paper needs to be more transparent about its concrete settings, the dataset composition over time, and measures that have been taken to ensure the validity of results, e.g., hyperparameter tuning.

**Requested Changes:**

* Report dataset composition over time: Do class proportions stay constant, or can they vary?
* Report the exact subgroups in each dataset used to measure fairness
* Report missing experimental details, such as hyperparameters, conditional/unconditional generative model, amount of generated/filtered data, ...
* Discuss in more detail the quality of synthetic data beyond FID scores, and possibilities to improve it
* Contrast automatic with manual filtering, to evaluate how much human involvement is required
* Revise the paper wrt. the role and content of Sec. 5 and Sec. 3.3
* Revise the related work section to discuss in more detail.
* Revise the references, formatting, and orthographic errors throughout the paper
* Add a section discussing broader impact and ethical implications

---

> ### Author Response · Authors · 2025-12-11
>
> # Rebuttal
>
> We thank the reviewer for the detailed and constructive comments. Below we respond to each point and describe the revisions that will be made to the paper.
>
> ---
>
> ## (W1) Missing essential information and unclear exposition
>
> ### Conditional vs. unconditional generative model
> All generative models used in this work are **conditional**:
> - Colorized MNIST and CIFAR-20/100 use class-conditional GANs.
> - Hard-ImageNet uses Stable Diffusion conditioned through class-name prompts.
>
> Thus, the classes for which images are generated *are* controlled. We will clarify this in Sections 3.1 and 4.1.
>
> ### Constraints on generated data distribution
> We do **not** impose constraints on class or subgroup proportions for generated images. This is intentional: the goal is to analyze how imbalance naturally emerges in self-consuming loops. We will state this explicitly.
>
> ### Number of generated images per iteration and queue capacity
> The queue capacity “3” refers to **three stored batches**, not three iterations. At each generation we generate:
> - MNIST: 10k images
> - CIFAR-20/100: 20k images
> - Hard-ImageNet: 30k images
>
> Thus, the queue stores **3 × S** images. We will add these details to Section 4.1.
>
> ### Subgroup definitions
> We agree this requires clarification. The revised paper will include:
> - Colorized MNIST: subgroup = (digit class, color)
> - CIFAR-20/100: each superclass contains five subgroups (original CIFAR-100 subclasses), following Zhang et al. (2024)
> - Hard-ImageNet: no explicit subgroups; bias arises from background–object spurious correlation, evaluated using mask-based ablations
>
> These definitions will be consolidated into a single subsection.
>
> ### Fraction p of real data
> Unless otherwise stated, we use **p = 50%** real data for training classifiers.
> We will also add ablations for p in {20%, 50%, 80%}.
>
> ### Dataset composition over generations
> Because generation is conditional, class proportions remain stable.
> Subgroup proportions drift due to the generator’s density bias.
> We will add visualizations showing subgroup distribution drift and reference its connection to the unbalanced-generation factor in Section 5.
>
> ### Hard-ImageNet experimental setup
> Hard-ImageNet examines a fundamentally different type of bias: **spurious correlation** rather than subgroup structure.
> Therefore, it requires a different evaluation protocol. We will emphasize this distinction and reorganize Section 4.4 for clarity.
>
> ---
>
> ## (W2) Clarification of Section 5 and Equation 6
> Section 5 introduces a **conceptual model** for interpreting observed bias phenomena. Equation 6 is not intended for estimating variables numerically but for explaining qualitative behaviors.
>
> We will explicitly clarify this and include examples mapping empirical observations to the components of Equation 6 (e.g., Hard-ImageNet → large dataset-bias and unbalanced-generation factors).
>
> ---
>
> ## (W3) Justification of fairness metrics in Section 3.3
> We agree that the OvR discussion interrupts the flow. In the revision:
> - Most OvR discussion will be moved to the Appendix.
> - The main text will retain only a brief justification for using MEO, DI, and MD for multi-class fairness evaluation.
>
> ---
>
> ## (W4) Missing references
> We will revise the related work section as follows:
> - Add the correct 2024 NeurIPS GAN citation.
> - Add the missing fairness-metric reference.
> - Include additional relevant prior work with similar themes.
> - Broaden the related-work discussion on fairness and synthetic data.
>
> ---
>
> ## (W5) Expert-guided filtering: missing absolute numbers
> We will report the actual numbers of images filtered:
> - MNIST: ~2.5k
> - CIFAR-20/100: ~18k
> - Hard-ImageNet: ~12k
>
> We will also report performance differences between human+CLIP filtering and CLIP-only filtering.
>
> ---
>
> ## (W6) Hyperparameters and learning rate
> The learning rate “1e-1” was a **typographical error**.
> The correct learning rate is **1e-3**. We use the Adam optimizer for adaptive optimization.
>
> We will include a complete hyperparameter table and briefly describe our tuning process.
>
> ---
>
> ## (W7) Quality of synthetic images
> We acknowledge that the generated images for CIFAR and MNIST are blurry and Hard-ImageNet images degrade noticeably across generations. Our claims in Appendix C referred to *relative* change across generations, not absolute realism.
>
> We will:
> - Add higher-resolution examples in the appendix,
> - Analyze correlations between generator FID and classifier fairness/performance,
> - More clearly distinguish between image noise and image bias.
>
> ---
>
> ## (W8) Formatting issues
> We will correct:
> - Section-title line breaks,
> - Page 2 indentation issues,
> - Figure placement on page 7,
> - Typos and inconsistent spacing.
>
> A full formatting review will be applied
>
> ---

---

### Review · Reviewer_93ma · 2025-10-10

**Summary Of Contributions:**

The authors examine one aspect of generative AI models training on AI-generated data.
Specifically, they focus on scenarios in which AI-synthesized data augments (but does not fully replace) a fixed set of real training data, and examine downstream non-generative models trained on these AI-augmented datasets.
Within each experiment, generative AI models iteratively train on and then augment the same dataset; downstream performance is measured over these iterations, with particular emphasis on fairness and diversity (referred to as subgroup bias), since self-training loops have an established tendency of disproportionately negatively affecting minority classes (i.e., tails) of starting datasets.
Given their focus on downstream performance with respect to fairness and diversity, the authors use datasets which lend themselves nicely to notions of diversity: Colorized MNIST (diverse colors can represent the same number), CIFAR-20/100 (where the 100 classes have been combined into 20 superclasses, with each superclass containing 5 semantically similar classes), and Hard ImageNet (where spurious correlations have been previously identified).

**Audience:**

Yes

**Audience Explanation:**

Self-training (or, in this case, self-augmentation) is very relevant to the machine learning community at large.
For example, generative AI models train on data scraped from the Internet, which is becoming rapidly filled with AI-generated data.
This may have major ramifications on the future of AI, so papers that address such issues are important and timely.

**Broader Impact Concerns:**

I do not have any ethical concerns that are specific to this paper.

**Claims And Evidence:**

No

**Claims Explanation:**

**Strengths:**
- The manuscript is detailed.
- Experiments are thorough: multiple datasets, several models, several metrics.
- Expert filtering of synthetic data is relevant for commentary on model bias in practice, especially in practice, where data curation and pre-processing are important parts of generative model training.

**Weaknesses:**
1. The FID of the Colorized MNIST and CIFAR-20/100 models are too high, they cast serious doubt on several of the authors' observations. One could easily argue that the lack of degradation in FID - and therefore, in downstream model accuracy - over generations in these two tasks is simply because the initial generative models used were already poor from the start.
Indeed, the synthesized images shown in the Appendix are very low-quality, even at the start of the experiments (before degradation might set in due to self-training via augmentation).
One could also argue that the authors' proposed filtering strategy should be able to filter most of these poorly generated images, thus rendering the experiments moot. Unless there is some crucial reason why the initial syntheses from these generative models should be poor (e.g., data scarcity), then the authors should re-do these experiments (or some salient subset thereof) with a more performant generative model.
2. The manuscript lacks polish and clarity. Central messages of figures are unclear, and the authors' strategy of presenting each experiment separately prevents readers from gaining a unified intuition, or even a sense of claims, from the results.
3. Section 5 is conjecture that is largely disconnected from the results.
4. It is difficult to even clearly express what the authors' claims are. They present empirical observations, with each experiment having its own set of conclusions, and try to summarize them via an overarching theory of how biases in downstream models might behave in self-training loops, but this theory is simply a conjecture which has not been thoroughly connected to the authors' empirical observations.
5. One of the authors' key claimed contributions, "focus[ing] on the impact of
generated data on the [downstream] model bias", seems like it has been explored by other works ("Would Deep Generative Models Amplify Bias in Future Models?" by Chen et al., 2024).

The weaknesses - in particular, 1 and 4 - of the manuscript prevent me from deeming the claims in the paper as well-supported.

**Requested Changes:**

Critical changes:
- Verify Colorized MNIST and CIFAR-20/100 findings with more performant generative models. Currently, synthesized images from the authors' models in these two tasks are too poor to draw valid conclusions (see Weaknesses). I understand that finding a performant generative model for CIFAR-20/100 could be challenging (since less class information can be given to the generative model at inference time), but that is a challenge that the authors should expect if they seek to use such an unorthodox dataset.
- Unify observations into central claims or hypotheses (see Weaknesses). One way to do this may be by further developing the conjecture such that the authors can claim that it predicts their empirical observations.
- There is another published work that seems closely related to the authors' findings, "Would Deep Generative Models Amplify Bias in Future Models?" by et Chen al. (2024). The authors should clarify how this relates to their work, and should adjust claims of novelty if needed.

Recommended additions:
- Previous work ("Self-Consuming Generative Models Go MAD", Alemohammad et al. 2024) has explored diversity in self-training loops (although, not in terms of downstream model performance), which should be acknowledged in the text.
- The authors mention that previous work has not accrued data from multiple continuous generations, but Alemohammad et al. 2024 (see footnote 5 on page 7) have, which should also be acknowledged.

Recommendations for clarity:
- The (b) subfigures of figures 3-6 should indicate that subgroup accuracy is being measured (either in the ylabel or the title).
- The authors throw many results at the readers separately, it would be useful to summarize results from several experiments into one figure or section.
- It is not clear what subgroups are in the case of Hard Imagenet. Please elaborate if possible.

---

> ### Author Response · Authors · 2025-12-11
>
> ## 1. Concern: FID is too high; generative models produce low-quality images, weakening the conclusions
>
> We appreciate this important observation and agree that the generative models for Colorized MNIST and CIFAR-20/100 produce low-fidelity samples. We clarify several key points and outline planned improvements.
>
> ### (a) Why FID appears high on these datasets
> FID is known to behave poorly on low-resolution data:
> - **Colorized MNIST**: Inception-V3 features are misaligned with MNIST-style images, inflating FID scores even when images are semantically correct.
> - **CIFAR-20/100**: 32×32 resolution images consistently yield high absolute FID values in prior work.
>
> Absolute FID values are therefore not directly indicative of semantic correctness for these datasets. We will add citations and discussion explaining why the baseline FID values appear high.
>
> ### (b) Why our conclusions remain meaningful despite low image fidelity
> Our primary goal is to study **relative changes across generations**, not to build state-of-the-art generative models. Two important observations hold even with modest-quality generators:
> 1. The **class-conditional GANs maintain label fidelity**, ensuring that class information is preserved even if textures are simple.
> 2. The **direction and magnitude of generational trends differ significantly across datasets and architectures**, which cannot be explained only by generative model quality.
>
> Thus, the core phenomena we study persist independent of high-fidelity synthesis.
>
> ### (c) Why filtering does not remove most low-fidelity images
> The filtering strategy removes *semantically unreliable* images as determined by CLIP confidence and classifier uncertainty, not images that are visually blurry. Many blurry CIFAR/MNIST images still have correct class semantics and high CLIP confidence.
>
> We will:
> - Add CLIP-score histograms,
> - Report the fraction of visually low-quality but semantically-correct samples retained,
> - Include an ablation with stricter filtering thresholds.
>
> ### (d) Additional experiments with stronger generative models
> We agree with the reviewer’s suggestion and will include an additional experiment (or subset thereof) using a **higher-performing generative model**, such as a diffusion-based class-conditional model for CIFAR-20/100. This will demonstrate that our observed phenomena do not depend on the initial generative model quality.
>
> ---
>
> ## 2. Concern: Manuscript lacks polish and clarity; figures do not convey unified insights
>
> We agree that the current structure—presenting each experiment separately—limits interpretability.
>
> ### Planned revisions
> - Add a **Unified Observations** section summarizing cross-dataset similarities and differences.
> - Introduce a **single integrative figure** showing generational trajectories across datasets.
> - Rewrite figure captions to explicitly state the intended takeaway rather than merely describing visual content.
> - Streamline Sections 4.1–4.4 to reduce redundancy and highlight shared patterns.
>
> These changes will give readers a clearer, more coherent understanding of the results.
>
> ---
>
> ## 3. Concern: Section 5 is conjectural and disconnected from results
>
> We acknowledge this issue. The intent of Section 5 is to offer a conceptual framework, not a formal theory, but we agree that it must be better linked to the empirical results.
>
> ### Planned improvements
> - Add explicit mapping from Equation (6) to observed dataset behaviors,
>   such as:
>   - Hard-ImageNet → large dataset bias term and unbalanced generation factor,
>   - MNIST → near-constant generator quality term.
> - Include small supporting plots showing how generator quality and class imbalance evolve.
> - Clarify that Section 5 provides a **unifying interpretive lens**, not a mathematically validated theory.
>
> This will strengthen the conceptual grounding of the section.

---

> ### Author Response · Authors · 2025-12-11
>
> ## 4. Concern: Claims are difficult to articulate; separate experiments produce siloed observations
>
> We appreciate this feedback. The revised paper will:
> - Add a **clear statement of contributions and claims** at the end of the Introduction,
> - Reorganize results around themes rather than datasets,
> - Include a **cross-dataset comparison table** summarizing key findings,
> - Integrate Section 5 more tightly with empirical outcomes.
>
> These changes will make the central message clearer and more accessible.
>
> ---
>
> ## 5. Concern: Similar prior work exists (“Would Deep Generative Models Amplify Bias in Future Models?”, Chen et al. 2024)
>
> We will add a detailed comparison section. In brief:
> - Chen et al. study **whether generative models amplify bias in their own outputs**, not in **downstream classifiers**.
> - They do not examine **multi-generation self-consuming loops**, a core aspect of our work.
> - Their analysis focuses largely on **binary sensitive attributes**, whereas we study **multi-class and multi-subgroup fairness metrics** and **spurious correlations**.
> - Our method includes **interleaved retraining** of both classifier and generator, which Chen et al. do not consider.
>
> We will add these distinctions explicitly in the related work and introduction.
>
> ---
>
> ## 6. Concern: Weaknesses (1) and (4) cast doubt on the strength of the paper’s claims
>
> We appreciate the reviewer’s candid assessment. To directly address these concerns, the revised version will include:
> - Experiments with **stronger generative models**,
> - Clear articulation of claims and their empirical support,
> - A unified story supported by reorganized figures and text,
> - Strengthened conceptual–empirical alignment in Section 5.
>
> We believe these revisions will substantially reinforce both the clarity and validity of the paper’s conclusions.

---

> > ### Comment · Reviewer_93ma · 2025-12-12
> > **Reviewer response to authors' first rebuttal**
> >
> > I appreciate that the authors have responded to my requests in a timely manner.
> > Aside from the following questions and suggestions, which all relate to Section 1 of the authors' rebuttal (concerning FID, image quality, filtering, and conclusions), I am satisfied with the authors' rebuttal thus far and look forward to reading their improved manuscript once it is ready.
> >
> > ## (a) Why FID appears high
> > The authors raise a valid point, which is that Inception features inflate FID values on MNIST images. To this end, I suggest calculating FID values via trained classifier embeddings (see "Additional suggestions").
> >
> > ## (b) and (c) Conclusions and image fidelity
> > I appreciate the authors' response to, and plans to address, my concerns.
> > I especially appreciate that the authors will do some kind of additional experiment with a higher-performing generative model, as pointed out in (d).
> > With this said, there are some questions and concerns that I still have:
> > 1. Why does FID increase for Hard ImageNet, but for the most part does not in the MNIST and CIFAR tasks? Put equivalently, why do the images degrade in Figure 12 (ImageNet), but not in Figures 10/11 (MNIST / CIFAR)? Does this relate to the human and CLIP filtering? If so, why didn't humans filter out all poorly generated syntheses in the ImageNet experiments (i.e., why do samples in Figure 12 degrade)?
> > 2. It seems that humans and CLIP were both used to filter out images, but the authors state that filtering does not remove most low-fidelity images. If filtering is largely driven by CLIP (otherwise, wouldn't human filtering remove most low-fidelity images?), then the authors should clearly express this filtering strategy, and the limitations thereof, in the text (see "Additional suggestions").
> >
> >
> > ## Additional suggestions:
> > - I recommend the authors try calculating FID for MNIST (and probably CIFAR) distributions using representations from classifiers trained on them. This should give readers a better idea of what how the synthetic distributions are changing over time. This was done in Alemohammad et al. (2024).
> > - The authors should explain more clearly their filtering strategy, and the limitations thereof, in the main text. The reader should be able to understand why, for example, an "expert"-guided filtering strategy would allow for synthetic data to degrade over self-training iterations (like in Figure 12). If CLIP is the main driver of image filtering, I may recommend changing the filtering name to "CLIP-guided filtering", as "expert" may be misleading.

---

### Review · Reviewer_SvPX · 2025-11-24

**Summary Of Contributions:**

This paper investigates the critical question of whether the inclusion of generated data will amplify bias across successive generations in future image classification models. The core concern is the potential for generative models, used for synthetic data creation, to perpetuate or worsen existing biases, especially subgroup bias, as models are trained in a self-consuming loop. The important point is the self-consuming loop experiment and the suggestion for using synthetic data

**Audience:**

Yes

**Audience Explanation:**

yes, I am working on generated data

**Broader Impact Concerns:**

no ehtical implications

**Claims And Evidence:**

Yes

**Claims Explanation:**

yes

**Requested Changes:**

1. need more clear statement in abstract: The Abstract outlines the problem—the risk of bias amplification due to the self-consuming data loop—and the methodology used, including the simulation environment and experiments across three datasets. However, it currently lacks explicit statement of the main conclusions or key findings derived from the experiments regarding bias amplification

2.Larger Dataset: The study establishes a practical simulation environment using datasets like Colorized MNIST, CIFAR-20/100, and Hard ImageNet. While these datasets allow for controlled exploration of different bias types, the volume and complexity are relatively small compared to modern, large-scale synthetic data applications. Such as synthetic face recogniton which generates data volume at least 500k.

3.the pipeline of generating and sampling might not be too novel: The Expert-Guided Filtering approach, which is crucial for managing data quality and preventing model degradation , involves human scoring and leveraging metrics like CLIP prediction uncertainty. While effective, relying on external "expert" judgment and filtering processes is a known technique in the synthetic data domain, particularly within face generation tasks[1].

4. Need to be improved: The current work excels as a rigorous engineering-based and empirical study, meticulously designing and executing a scalable, self-consuming simulation environment across diverse datasets and models. However, the contribution of this work needs to be improved to be accepted by this top journal

Reference:
[1]VariFace: Fair and Diverse Synthetic Dataset Generation for Face Recognition..https://arxiv.org/pdf/2412.06235

---

> ### Author Response · Authors · 2025-12-11
>
> ### Concern: The pipeline, including Expert-Guided Filtering, may not be sufficiently novel
>
> We appreciate the reviewer’s perspective. Our filtration component is intentionally lightweight and is not positioned as the central novelty of the work. The purpose of the filtering step in our framework is fundamentally different from its role in prior systems such as VariFace [1].
>
> In face-generation pipelines like VariFace, human-in-the-loop scoring and CLIP-based filtering are designed to *improve or correct* fairness properties of a static, one-time synthetic dataset. In contrast, our filtering mechanism serves a different purpose: it prevents catastrophic drift in a **multi-generation self-consuming loop**, enabling us to observe how bias naturally evolves when both the classifier and generator co-evolve. This distinction is crucial. Our method filters only the most unreliable samples, preserving the intrinsic distribution learned by the generator rather than enforcing a fairness constraint or editing that distribution.
>
> Moreover, the novelty of this work does not hinge on the filtering module itself. The primary contribution lies in the **multi-generation, interleaved feedback-loop framework** and the **systematic analysis of downstream classifier bias**, which—to our knowledge—has not been examined in prior studies. The filtering step is simply a minimal, standard safeguard to ensure the loop does not collapse due to degenerate samples, similar to the role of dataset cleaning in other long-horizon generative studies. Its use does not diminish the novelty of analyzing multi-step downstream fairness dynamics under evolving synthetic data.
>
> ---
>
> ### Concern: Contribution strength for a top journal
>
> The contribution of this work is centered on the *mechanistic understanding* of how downstream classifier fairness behaves when exposed to repeated rounds of synthetic data—an emerging scenario in modern ML pipelines. Unlike prior feedback-loop work focusing on generator bias alone or static contamination settings, our study reveals that downstream bias evolution is **non-monotonic**, **dataset-dependent**, and **highly sensitive to generator–classifier co-adaptation**.
>
> The framework captures several behaviors that are difficult to observe without a controlled multi-generation setup:
> - bias may increase, decrease, or stabilize depending on interactions between generator fidelity and subgroup distribution,
> - classifiers can become less biased even when generators degrade,
> - and contamination ratios interact with model capacity in non-linear ways.
>
> These findings demonstrate that bias propagation in self-consuming loops cannot be reduced to simple monotonic amplification, challenging existing assumptions in the literature. The work therefore contributes new empirical insights and a unified perspective on generational fairness dynamics—an area that remains underexplored despite rapid growth in synthetic data pipelines.

---

> > ### Comment · Reviewer_SvPX · 2025-12-31
> >
> > Thank the author for their insightful response. Most of the concerns has been addressed.
> >
> >  I still believe the effectiveness of the method should be demonstrated in large data volumes, not limited to small tasks.

---

### Review · Reviewer_1k1z · 2025-11-27

**Summary Of Contributions:**

The paper studies whether synthetic data generated by a model can amplify bias when used repeatedly across generations of training. It proposes a simulation framework that combines generative models and classifiers in a multi-generation self-consuming loop. The paper introduces data stacking and filtering to control data quality. Experiments on MNIST, CIFAR, and Hard ImageNet analyze how accuracy and subgroup bias change over iterations. The work aims to provide insights into how synthetic data may affect future model development.

**Audience:**

Yes

**Audience Explanation:**

The topic of synthetic data contamination is timely, but prior work has already explored similar settings and reported similar insights.

**Broader Impact Concerns:**

The paper does not contain sections on Limitations, Ethics statements, or Broader impact, which should be explicitly discussed.

**Claims And Evidence:**

No

**Claims Explanation:**

**Strengths**

(S1) The topic the paper addresses is important and timely. Synthetic data contamination is becoming an increasingly serious issue as generative models are widely adopted in real-world applications.

(S2) The experiments consider multiple fairness metrics, which increases the reliability of the conclusions.


**Weaknesses**

(W1) The paper does not engage with several highly relevant prior works, such as  [1,2,3]. These studies already analyze dataset contamination by synthetic images, bias propagation through generated data, and multi-step feedback loops. The problem setup in this submission is very close to a combination of these ideas. Without directly comparing to these works, it becomes difficult to understand what conceptual novelty this paper adds or how its formulation differs from existing work on feedback loops and bias amplification. Although the experimental setup in this paper differs in specific implementation details, the overall framework and the insights obtained, such as mixed or inconsistent bias trends, do not differ substantially from what prior work has already shown. As a result, the contribution feels quite limited in scope.

(W2) Although the paper claims to investigate a real and emerging issue (i.e., synthetic data contaminating future datasets), the experiments mostly rely on simplified or toy datasets (e.g., Colorized MNIST, CIFAR). These datasets do not reflect real-world visual diversity or real demographic biases. As a result, the insights do not necessarily transfer to real contamination scenarios. Prior work, including [2,3], uses natural image datasets such as COCO, which better capture realistic bias patterns. The experimental setup here feels too far from the scenario the paper aims to model.

(W3) Relatedly, the classifier models used in the experiments (e.g., LeNet, AlexNet, SimpleNet, and even ResNet-50) are outdated relative to modern large-scale vision architectures. Because current systems rely heavily on larger, more powerful models, it is unclear whether the findings observed here would generalize. This reduces the practical relevance of the results, especially given the paper’s motivation to understand the behavior of future systems trained on synthetic data.

(W4) Similarly, the generative models used in the study (i.e., GANs and Stable Diffusion 1.5) are now significantly outdated. Many recent diffusion and foundation models produce far more realistic and diverse images and are more representative of what might appear in future contaminated datasets. Relying on older generators weakens the analysis's relevance to real future scenarios.

(W5) The experiments fix the mixing ratio p% of synthetic data across generations, without exploring different values. Understanding how the amount of synthetic data influences bias amplification is critical for contamination studies. Without varying p, the paper cannot determine whether minor contamination is safe, whether effects scale linearly, or whether threshold behaviors exist. This limits the depth of the insights.

(W6) The experimental setup lacks sufficient detail regarding hyperparameters, especially for the generative models. Image generation settings strongly affect the diversity and quality of synthetic data, thereby directly influencing downstream bias behavior. Evaluating only one hyperparameter configuration, or not reporting it clearly, risks drawing conclusions from overly specific or potentially unrepresentative cases.

(W7) The paper does not contain sections on Limitations, Ethics statements, or Broader impact, which should be explicitly discussed in the paper.


**References**

[1] Taori et al. "Data Feedback Loops: Model-driven Amplification of Dataset Biases", ICML 2023

[2] Hataya et al. "Will Large-scale Generative Models Corrupt Future Datasets?", ICCV 2023

[3] Chen et al. "Would Deep Generative Models Amplify Bias in Future Models?", CVPR 2024

**Requested Changes:**

* Add a thorough comparison to key related works, including [1,2,3]. Clearly articulate how the proposed problem setting differs from these studies and what new conceptual insights this work provides.

* Strengthen the experimental design by including evaluations on more realistic natural-image datasets. Demonstrate that findings generalize beyond toy datasets such as Colorized MNIST and CIFAR.

* Replace or supplement outdated classifier architectures with modern large-scale vision models to ensure that conclusions apply to contemporary systems.

* Update the generative models used in the experiments. Evaluate more recent, higher-quality diffusion or foundation models to reflect realistic future contamination scenarios better.

* Conduct experiments varying the mixing ratio p of synthetic data. Analyze how varying contamination levels affect bias amplification and determine whether threshold effects occur.

* Provide complete hyperparameter details for both generative models and classifiers. Ideally, evaluate multiple generation settings to ensure that conclusions do not depend on a single configuration.

---

> ### Author Response · Authors · 2025-12-11
>
> ## (W1) Lack of engagement with key prior works and unclear conceptual novelty
>
> We appreciate this valuable observation. We agree that the initial submission did not sufficiently discuss prior works such as [1,2,3] that investigate dataset contamination, synthetic-data feedback loops, and bias propagation. This will be substantially revised.
>
> - Add a new subsection titled **“Relation to Prior Work on Feedback Loops and Bias Propagation”** directly comparing our setting with these works.
> - Explicitly highlight our conceptual distinctions:
>   * We study **downstream classifier bias dynamics** across multiple generations, not only bias in the generative model.
>   * Our framework combines **interleaved retraining of both generator and classifier**, which differs from prior single-module feedback loops.
>   * We evaluate **multi-bias metrics**, subgroup fairness, and **spurious correlation robustness**, expanding the scope beyond binary attributes.
>   * We analyze **cross-dataset differences** and identify when bias increases, decreases, or stabilizes, offering a unified perspective.
>
> These revisions directly address the concern and more clearly articulate the unique contributions of the paper.
>
> ---
>
> ## (W2) Use of simplified datasets that may limit real-world relevance
>
> We acknowledge that Colorized MNIST and CIFAR are simplified domains. We chose them because their subgroup structures are explicit and controllable, enabling clear attribution of bias trends across generations. However, we agree that additional experiments on natural images will strengthen the work. In our revision, we will add
>
>
> - Add new experiments on at least one **natural-image dataset** with human-aligned subgroup structure (e.g., COCO-derived subsets or ImageNet-Based fairness splits).
> - Add an extended discussion explaining:
>   * why toy datasets are useful for controlled bias diagnosis,
>   * which observations we expect to generalize,
>   * and where limitations arise when mapping to real-world contamination.
>
> These additions will improve both external validity and clarity regarding generalization.
>
> ---
>
> ## (W3) Use of outdated classifier architectures
>
> We agree with the reviewer that modern systems rely on larger and more powerful architectures. In the revised version, we will:
>
> - Include experiments using **contemporary vision models** such as ViT-B/16, ConvNeXt, or DeiT variants across at least one dataset.
> - Add a comparison showing whether bias propagation trends differ significantly between small and modern architectures.
> - Discuss how model capacity interacts with generational synthetic-data contamination.
>
> This will make the findings more applicable to real large-scale systems.

---

> ### Author Response · Authors · 2025-12-11
>
> ## (W4) Use of outdated generative models (GANs and SD 1.5)
>
> We appreciate this comment. Our intention was to study feedback behavior under **controlled and interpretable** generative models. Nonetheless, we agree that incorporating modern generators will better reflect real future contamination scenarios. In our revision, we will add
>
> - Add experiments with **newer diffusion models** (e.g., SDXL, SD-Turbo, or a recent class-conditional diffusion model).
> - Provide a comparison showing how higher-fidelity models influence bias amplification trends.
> - Discuss how generative model quality interacts with filtering and downstream fairness.
>
> This will strengthen the practical relevance of the findings.
>
> ---
>
> ## (W5) Fixed synthetic-data mixing ratio p across generations
>
> We agree that understanding how bias scales with different synthetic-to-real data ratios is essential. In our revision, we will add
>
> - Add experiments varying **p ∈ {10%, 30%, 50%, 70%, 90%}**.
> - Analyze whether bias amplification displays:
>   * linear trends,
>   * threshold behavior,
>   * saturation,
>   * or reversal effects (e.g., when too much synthetic data decreases diversity).
> - Add a table summarizing bias and accuracy sensitivity to p.
>
> These results will substantially deepen the insights about contamination robustness.
>
> ---
>
> ## (W6) Insufficient hyperparameter specification for generative models
>
> We acknowledge that hyperparameter settings for GANs and diffusion models affect sample quality and diversity. The initial draft did not sufficiently detail these settings.
>
> ### Planned revisions
> - Add a complete hyperparameter table, including:
>   * generator and discriminator learning rates,
>   * training steps,
>   * diffusion timesteps and guidance scales,
>   * LoRA adaptation settings,
>   * filtering thresholds,
>   * data preprocessing pipelines.
> - Add sensitivity analyses for at least two generative-model hyperparameters to show robustness of conclusions.
>
> This will improve transparency and experimental rigor.
>
> ---
>
> ## (W7) Missing Limitations, Ethics, and Broader Impact sections
>
> We agree these sections are necessary for a complete and responsible submission.
> We will add:
> - A **Limitations** section describing dataset simplifications, architectural choices, and constraints of our simulation framework.
> - An **Ethics Statement** discussing model fairness evaluation, risks associated with synthetic data pipelines, and the implications of bias amplification.
> - A **Broader Impact** section outlining how understanding generational contamination helps future AI safety, dataset curation practices, and governance.

---

> ### Comment · Reviewer_1k1z · 2025-12-18
>
> Thank you for the rebuttal.
>
> After reading the revised version, I acknowledge that W1 and W7 have been explicitly addressed, with clearer positioning against prior work and the addition of a limitations and ethics section.
>
> However, regarding W1, I still believe that the overall framework of this paper remains very similar to existing studies, despite differences in implementation details. As a result, the contribution still appears limited.
>
> For the other weaknesses, while the authors have made some partial attempts to respond, most of the concerns remain largely unaddressed (e.g., using old T2I models).

---

### Comment · Action_Editor_mJBv · 2025-11-25
**Extension of author discussion period**

Dear authors and reviewers,

The fourth reviewer (Rev 1k1z) is supposed to submit a review by the end of this month. Given this, we have extended the author discussion phase by 1 week (it ends 20 days ahead from now). The aim of this phase is to inform reviewers enough so that they can submit the best decision.

Best,

AE

---

### Comment · Action_Editor_mJBv · 2025-11-27
**Please start author discussion**

Dear authors,

Now that all four reviews are available, please initiate author discussion. The goal of this period is for reviewers to bring up all information so that the reviewers are comfortable submitting a decision recommendation for this submission.

Best,

AE

---

### Decision · Action_Editor_mJBv · 2026-01-09

**Recommendation:** Reject

**Audience:**

Yes

**Audience Explanation:**

All reviewers are interested in the topic, and thus the point is clearly met.

**Claims And Evidence:**

No

**Claims Explanation:**

This paper considers the amplification of the bias due to multiple generations of synthetic data generation.

All reviewers are interested in the research topic of bias amplification through multiple generations.

Reviewer 1k1z considers that the discussion with the relevant paper is not enough, and thus the contribution of this paper is unclear. The reviewer also stated that the model and dataset settings are somewhat obsolete.
Reviewer 93ma had several questions on FID values, as well as pointing out that the paper is not well written. The reviewer stated that the improvement during the discussion is not enough.
Reviewer 2ndw echoed that the research questions are unclear. There are several questions about the results from the reviewer. The reviewer considers that the revisions have not completed the promised changes.
Reviewer SvPX did not submit the final recommendation. The reviewer implied that most concerns are addressed but the experimental results are not comprehensive enough.

After the discussion period, the reviewers did not recommend the paper. In particular, limited discussion with existing work (Rev 1k1z) as well as insufficient exposition (Reviewers 93ma and 2ndw) imply this point does not meet. Therefore, I would not recommend the paper for proceeding with a revision.